# Efficient consideration of coordinated water molecules improves computational protein-protein and protein-ligand docking discrimination

**Ryan E. Pavlovicz**[1,2¤], **Hahnbeom Park**[1,2], **Frank DiMaio**[1,2]*

**1** Department of Biochemistry, University of Washington, Seattle, Washington, United States of America,
**2** Institute for Protein Design, University of Washington, Seattle, Washington, United States of America

¤ Current address: Cyrus Biotechnology, Seattle, Washington, United States of America
* dimaio@u.washington.edu

**Data Availability Statement:** Native water sets are available as Supporting Information Files. We have uploaded the data set into a public git repository

## Abstract

Highly coordinated water molecules are frequently an integral part of protein-protein and protein-ligand interfaces. We introduce an updated energy model that efficiently captures the energetic effects of these ordered water molecules on the surfaces of proteins. A two-stage method is developed in which polar groups arranged in geometries suitable for water placement are first identified, then a modified Monte Carlo simulation allows highly coordinated waters to be placed on the surface of a protein while simultaneously sampling amino acid side chain orientations. This "semi-explicit" water model is implemented in Rosetta and is suitable for both structure prediction and protein design. We show that our new approach and energy model yield significant improvements in native structure recovery of protein-protein and protein-ligand docking discrimination tests.

## Author summary

Well-coordinated water molecules—those forming multiple hydrogen bonds with nearby polar groups—play an important role in the structure of biomolecular systems, yet the effect of these waters is often not considered in molecular energy computations. In this paper, we describe a method to efficiently consider these water molecules both implicitly and explicitly at the interfaces formed by two polar molecules. In computations related to determining how a protein interacts with binding partners, we show that the use of this new method significantly improves results. Future application of this approach may improve the design of new protein and small molecule drugs.

This is a *PLOS Computational Biology* Methods paper.

which may be found at https://github.com/
rpavlovicz/rpavlovicz-docking_data_sets.

**Funding:** Funding for this research was provided
by NIH General Medical Sciences award
(GM123089) to FD. The funders had no role in
study design, data collection and analysis, decision
to publish, or preparation of the manuscript.

**Competing interests:** RP is employed at Cyrus
Biotechnology with granted stock options. Cyrus
Biotechnology distributes the Rosetta software.

## Introduction

Water plays a significant role in biomolecular structure. The hydrophobic effect drives the collapse of proteins into their general shape while highly coordinated water molecules (water molecules making multiple water-protein hydrogen bonds) on the surface of a protein may confer specific conformations to nearby polar groups. Furthermore, water plays a key role in biomolecular recognition: when a ligand binds its host in an aqueous environment, it must displace water molecules on the surface and energetically compensate for the lost interactions [1]. Coordinated water molecules may also drive host-ligand recognition by bridging interactions between polar groups on each side of the complex.

Simulations of proteins in explicit solvent have been successful in predicting folded conformations[2] as well as computing binding free energies[3] with high accuracy. Explicit solvent calculations are computationally expensive, particularly in Monte Carlo simulations where a long water equilibration period might be required. Such a cost may be alleviated through the use of an implicit solvent[4] model, which while more efficient, incurs a loss of accuracy by disregarding the energetics of highly-coordinated water molecules[5]. Thus, an approach combining the efficiency of implicit solvation with the ability to recapitulate well-coordinated water molecules is desired. Several such methods have been developed but tend to be developed for specific types of interactions (eg. protein-protein or protein-small molecule ligand) [6–11] or are computationally expensive[12].

In this paper, we describe the development of general methods for capturing the energetic effects of explicit solvent, but with the computational efficiency of an implicit solvent model. Our intent is that this energy model is better at discriminating the correct binding modes of protein-protein and protein-ligand complexes. These new methods include: 1.) a new energy function that implicitly captures the energetics of protein and coordinated-water interactions and 2.) a conformational sampling approach that efficiently samples protein and explicit water conformations simultaneously. We show that these methods enable us to predict water positions accurately, as well as improving our ability to discriminate native protein-protein and protein-ligand interfaces from non-native decoy conformations.

## Results

Our approach for modeling coordinated water molecules using Rosetta, fully described in *Methods*, is briefly presented here. We have developed two complimentary approaches for capturing coordinated-water energetics. We hypothesize that more accurately modeled interface waters will lead to better discrimination of correct binding modes from incorrect (decoy) binding modes. First, *Rosetta-ICO (Implicit Consideration of cOordinated water)*, implicitly captures pairs of polar groups arranged such that a theoretical "bridging" water molecule may form favorable hydrogen bonds to stabilize the interaction. This calculation is efficient but ignores multi-body interactions that may favor, for example, waters coordinated by >2 hydrogen bond donors or acceptors. While this implicit water model is more accurate than our prior model, which did not consider these water molecules at all, modeling a subset of waters explicitly should further improve model accuracy. Therefore, we have also developed *Rosetta-ECO* (*Explicit Consideration of cOordinated water*), in which Rosetta's Monte Carlo (MC) simulation is augmented with moves to add or remove explicit solvent molecules from bulk. By sampling water orientations at sites where predicted bridging waters overlap (Fig 1E), we properly coordinate water molecules to optimize hydrogen bonding.

For both approaches, the Rosetta energy function has been reoptimized using the *dualOptE* framework described by Park et al.[14]. In this optimization, several meta-parameters describing the shape of the *Rosetta-ICO* potential; several terms controlling the strength and shape of

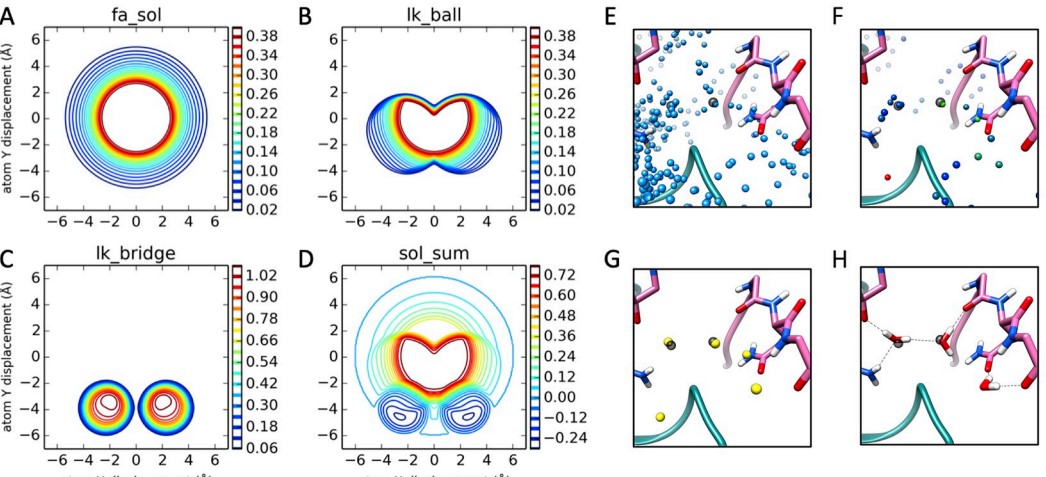

**Fig 1. Implicit and explicit treatment of water In Rosetta. Implicit water score function potentials, panels A-D.** Potential plots were generated by orienting the N-H and C = O groups of two ALA residues along the same axis with a H—O distance of 1.3 Å (origin). The donor residue is then shifted +/- 7 Å to generate a planar cut of the solvation potentials between the N and O atoms. All plots have units of kcal/mol[13, 14]. **(A)** fa_sol term: isotropic desolvation penalty implemented in Rosetta using the Lazaridis-Karplus model. **(B)** lk_ball term: anisotropic correction for polar atom types, first introduced into the REF2015 score function. **(C)** lk_bridge term: anisotropic solvation reward introduced into the *Rosetta-ICO* score function. **(D)** Composite of panels A-C, using the finalized *Rosetta-ICO* score term weights. **Explicit water placement with *Rosetta-ECO*, Panels E-H. (E)** Initial possible solvation sites (blue) are based on statistics of water positions around backbone polar atoms in addition to sites around side chain polar atoms considering all possible non-clashing rotamers. Pictured is the interface of PDB ID 1P57, between the N-terminal (pink) and catalytic (teal) domains of hepsin, with crystallographic waters in transparent grey. **(F)** After an initial stage of Monte Carlo packing of both the possible water sites and surrounding protein side chains, a cutoff is applied based on the water occupancy of each site over the simulation (blue = 0% occupancy, green = 25%, red = 50%). **(G)** Remaining water sites are clustered, and a second cumulative dwell time cutoff is applied. **(H)** The final predicted water sites are converted into three-atom water molecules and the orientation is reoptimized together with nearby sidechain conformations using the Rosetta all-atom energy function.

protein-water interactions; and ~50 other per-atom polar parameters were optimized to allow for compensating changes to the new energy terms. Energy function parameters for polar groups, including partial atomic charges, were refit using the same training tasks originally used in the parameterization of the *opt-nov15* energy function[14], now called *REF2015*[13]. While all parameters were optimized for *Rosetta-ICO* (see S7–S10 Tables in S1 Text for final values), only a subset of water-specific parameters were refit when developing the explicit water terms for *Rosetta-ECO*. The results in this section are shown with the updated energy functions compared to baseline tests run using the *REF2015* energy function[14].

## Rotamer and water recovery at protein-protein interfaces

A set of 123 native protein-protein interfaces from high-resolution X-ray crystal structures was used to test how well the new energy models perform at simultaneously predicting amino acid side chain conformations and coordinated water sites (data set details may be found in S2 Text). Tests involved the re-sampling of side chain conformations of interface residues on a fixed backbone in MC simulations and evaluating resulting predicted side chains against the deposited density maps. In tests involving semi-explicit water molecules (*Rosetta-ECO*), we *simultaneously* sample protein side chain conformations and water placements. A baseline rotamer recovery error of 9.73 ± 0.13% was obtained using the *REF2015* energy function for the 7040 flexible side chains of the test set. A marginal improvement is made with *Rosetta-ICO*, reducing error to 9.52 ± 0.04%. Inclusion of explicit water molecules in this test fails to

further decrease the overall rotamer recovery error beyond the improvements observed with *Rosetta-ICO*, with a *Rosetta-ECO* error of 9.59 ± 0.15%, while predicting ~19 explicit water molecules per protein-protein interface. For reference, side chain packing tests that use "native" water molecules (this would be the result of perfect water recall and precision) achieves a rotamer side chain recovery error of 8.36 ± 0.04%, while random perturbation of these waters suggest a placement tolerance of less than 0.8 Å (S17 Fig).

In addition to measuring side chain rotamer recovery at the protein-protein interfaces, we also analyzed the recovery of water positions found in the high-resolution X-ray crystal structures when implementing the *Rosetta-ECO* solvation method. For water recovery tests, modeled water positions are considered "correct" if they are placed within 0.5 Å of the native water or if they are coordinated by the same polar atoms. Using this strict criteria, *Rosetta-ECO* is able to recover 17.7% of native water molecules with a precision of 17.7%. Details of *Rosetta-ECO* water recovery are shown in Table 1. These data show that our approach is most effective at predicting "buried" waters (28.3% recovery) and highly coordinated waters (31.8% recovery of triply coordinated waters). Unsurprisingly, *Rosetta-ECO* is also much more effective at predicted backbone-coordinated waters, correctly predicting 50.0% of backbone-only coordinated waters. An example of two correctly predicted water sites is illustrated in Fig 1H.

*Rosetta-ECO* predicts an average 9.1 waters per 1000 Å$^2$, compared to an observed average of 9.3 waters per 1000 Å$^2$ of interface surface area, which is in line with previous analyses of interface solvation[15]. Thus, the *ECO* protocol hydrates protein interfaces to a similar degree as to what has been observed in crystallographic structures.

Finally, the results of *Rosetta-ECO* were compared against solvent placement using the 3D-RISM methodology as implemented in AmberTools19[16]. 3D-RISM, like most other water site prediction methods, operates on a fixed protein model (in this case, the crystallographic structures). In our tests, 3D-RISM recovered 22.9% of the full interface water data set, ~5% more than *ECO* when calibrated to the same level of precision (See S2 Table for detailed results). *Rosetta-ECO*, which predicts water positions *in addition* to protein side chain

**Table 1. Classification of predicted native waters (test set of 123).**

| | | *Rosetta-ECO* | |
|---|---|---|---|
| Type[1] | Subset Size | % recovered[2] | % precision[3] |
| All | 2815 | 17.7 (0.08) | 17.7 (0.08) |
| Exposed | 630 | 6.0 (0.13) | 4.7 (0.1) |
| Partially Buried | 1803 | 19.5 (0.39) | 21.7 (0.5) |
| Buried | 382 | 28.3 (1.19) | 27.5 (1.3) |
| 1 protein coord | 770 | 6.3 (0.12) | 5.0 (0.2) |
| 2 protein coord | 1077 | 27.2 (0.24) | 25.3 (0.3) |
| 3 protein coord | 399 | 31.8 (0.43) | 26.2 (0.4) |
| BB only | 330 | 50.0 (1.24) | 23.1 (0.4) |
| SC only | 333 | 7.8 (0.65) | 18.1 (1.1) |
| BB+SC | 440 | 27.6 (0.18) | 26.6 (0.3) |

[1]Three groups of categorization of type of predicted water molecules. First, waters are classified 'buriedness' based on number of amino acid neighbors (nCβ) with Cβ within 10 Å. Exposed: nCβ < = 15; partially buried: 15 < nCβ < = 25; buried: nCβ > 25. Second, classification by 1, 2, or 3 protein coordination partners within 3.2 Å. Finally, by type of coordinating protein atoms with 3.2 Å of the water O atom: at least two backbone only (BB only), side chain only (SC only) or a mix of backbone and side chain coordination (BB+SC).

[2-3]Percent of specific types of waters recovered using recovery criteria described in *Methods*, averaged over three runs with standard deviations in parentheses.

conformations, performs particularly strongly at recovering waters that are exclusively coordinated by backbone groups (Table 1), outperforming 3D-RISM by 35% for this classification of water. Overall, the 3D-RISM calculations take ~20-fold longer to run (S3 Table).

While other computational water predictions methods exist that are faster or more accurate than *Rosetta-ECO*, to our knowledge, they are all benchmarked against static protein structures, making direct comparison to *Rosetta-ECO* inappropriate. For example, WaterDock[9], which uses AutoDock Vina to predict water positions using a grid-based docking approach, was developed for computational drug design purposes, with a focus on small molecule binding sites as opposed to large protein-protein interfaces. Using a recovery cutoff of 1.4 Å, WaterDock reports a recovery of 87% of crystallographic waters to a set of 14 OppA crystal structures bound to different KXK tripeptides, with runtime on the order of seconds. Water-Map[11], on the other hand, relies on 2 ns MD simulations on a fixed protein. This leads to significantly longer run times (on the order of hours), but can yield highly accurate results: in a study using a dataset of 41 crystallographic water sites at nine bromodomain/ligand complex interfaces, WaterMap accurately predicted more than 70% of the experimental water positions within 0.5 Å[17]. 3D-RISM, which was also benchmarked in this study, recovered slightly more than 30% using the same recovery cutoff.

Finally, we also applied *Rosetta-ECO* to CAPRI Target 47[18], a homology modeling challenge of a protein/protein interface including the blind prediction of water molecules at the modeled interface. Our results, described in detail in S3 Text, places our best modeling effort with within range of the top-scoring submissions to the modeling challenge. One of our models places 13 water molecules at the modeled protein/protein interface, 11 of which come within 2.0 Å of one of the 22 crystallographic interface water molecules, making for a true positive prediction rate of 50% while only placing two additional water molecules not observed in the crystal structure.

## Native interface recapitulation

We next tested the ability the new energy model to recapitulate near-native conformations of protein-protein interfaces (PPIs) and protein-ligand interfaces. In these tests, which were not used at all in parameter training, the binding free energies for a number of near-native and incorrect (decoy) docking conformations of each complex are computed with the aim of discriminating the correct binding poses from the decoys. PPI decoys were sampled using a combination of Zdock[19] and RosettaDock[20], while protein-ligand decoys were generated using RosettaLigand[21]. Both datasets were enriched for water-rich interfaces, leading to 53 protein-protein and 46 protein-ligand interface datasets. Predicted interface energies, $\Delta G_{bind}$, were calculated for all decoys as described in *Methods*. We assess the ability to predict the near-native conformations using a "discrimination score,"[14] which computes the Boltzmann weight of near-native structures. The values range from 0 to 1, with higher values showing better discrimination. We also assess with a noisier (but more interpretable) "percent correct" metric, which identifies the number of cases in which near-native bound conformations have lowest energy. An overview of the results is shown in Table 2, while results for all cases are presented in S2 Fig through S10 Fig. Select cases in which the inclusion of predicted explicit water molecules improved native discrimination are detailed below.

## Protein-protein docking discrimination

In protein-protein docking discrimination tests with binding modes that broadly sample conformational space[14], significant improvements are observed when comparing *Rosetta-ICO* to the baseline results, with the discrimination score increasing from 0.63 to 0.74. *Rosetta-ECO*

**Table 2. Performance of solvation schemes on protein-protein and protein-small molecule docking discrimination.**

| | REF2105 | Rosetta-ICO[1] | Rosetta-ECO[2] |
|---|---|---|---|
| *Protein-small molecule* | | | |
| discrimination score[3] | 0.749 ± 0.003 | 0.807 ± 0.002 | 0.873 ± 0.003 |
| percent correct[4] | 77.1 ± 2.1 | 77.8 ± 1.8 | 94.1 ± 1.1 |
| run time[5] | 1.00 | 1.09 | 1.52 |
| Protein-protein | | | |
| discrimination score | 0.628 ± 0.014 | 0.739 ± 0.006 | 0.794 ± 0.004 |
| percent correct | 63.6 ± 0.9 | 74.9 ± 0.9 | 79.9 ± 2.3 |
| normalized run time | 1.00 | 1.25 | 2.59 |

[1]Implicit consideration of coordinated water molecules

[2]Inclusion of well-ordered explicit water molecules

[3]Reported are the average Boltzmann-weighted discrimination scores ± 1σ averaged over three independent runs for 46 protein-ligand and 53 protein-protein docking cases

[4]The percentage of cases in which the lowest scoring model is within 1.0 Å of the native conformation for protein-ligand docking and 2.0 Å for protein-protein docking, averaged over 3 independent runs

[5]Run time, normalized to baseline, is the sum of individual run times to calculate $\Delta G_{bind}$ for each near-native and decoy conformation

further improves this discrimination score to 0.79. We also consider the "success rate," the time the lowest-energy conformation is within 2.0 Å of native: the *ECO* model enables successful prediction of a near-native conformation in 8 additional cases out of the set of 53, a ~15% improvement. This comes at a modest increase in computational cost, with an average 1.25- and 2.59-fold increase in runtime for *ICO* and *ECO*, respectively.

As illustrated in Fig 2A, *Rosetta-ECO* improves the discrimination score for 38 of 53 cases, adding 13.4 water molecules to the average bound state and 15.0 water molecules to the average unbound state. These average improvements remain statistically significant. Looking at one such case (adrenodoxin reductase/adrenodoxin, PDB ID 1E6E), we see that while all three energy models correctly predict a near-native conformation, the "energy gap" between native and non-native conformations is improved under *Rosetta-ECO* (Fig 2B). Closer investigation of the near-native models shows 21 explicit water molecules added to the binding interface. The combined electrostatic and hydrogen bond energy contributions compose a large proportion of the improved binding energy, 5.2 kcal/mol more favorable than *Rosetta-ICO* for this particular binding configuration.

## Protein-ligand docking discrimination

For protein-ligand docking discrimination tests, *Rosetta-ICO* again shows an improvement over *REF2015*, with average discrimination score increasing from 0.75 to 0.81. *Rosetta-ECO* further increases the discrimination score to 0.87. In terms of "success rate", we see the same trend as with PPIs: *Rosetta-ECO* enables the correct prediction (within 1.0Å of native) in 7 additional cases out of the 46. These results indicate that both *Rosetta-ICO* and *ECO* help discriminate distant decoys from native conformations when compared to the *REF2015* energy model, with the inclusion of explicit water modeling in *ECO* conferring the largest benefit. This also comes at only a modest increase in run time: about 10% increased time for *ICO*, and about 52% increased computation time for *ECO*.

The improvements in discrimination score on a per-case basis are illustrated in Fig 3A. Here, we see that *Rosetta-ECO* provides a nearly across-the-board improvement in native discrimination compared to the baseline calculations. The individual energy distributions for

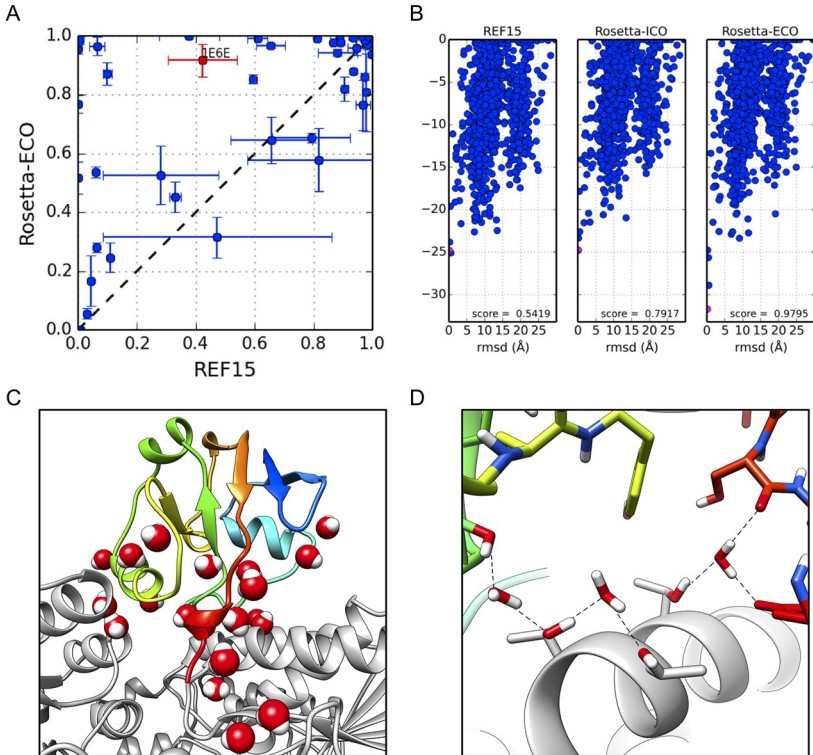

**Fig 2. Protein-protein docking results. (A)** Scatter plot comparing results of 53 cases between *REF2015* and *Rosetta-ECO*. Values are the average Boltzmann-weighted discrimination score ± 1σ from three independent runs. **(B)** Energy funnels for PDB ID 1E6E, adrenodoxin reductase bound to adrenodoxin (red data point in 2A), plotting computed $\Delta G_{bind}$ vs. RMSD from the native binding conformation for three different scoring methods. Discrimination scores for each distribution are noted in bottom right of each plot. **(C)** Explicitly solvated near-native docking pose (RMSD = 0.14 Å; pink data point in 2B) with the reductase in grey and adrenodoxin in rainbow (N- to C-terminus colored blue to red). **(D)** Coordination of some predicted interface waters.

PDB ID 1X8X (tyrosyl t-RNA synthase / tyrosine) in Fig 3B show how both *REF2015* and *Rosetta-ICO* incorrectly favor a decoy 6.6 Å from native. *Rosetta-ECO*'s explicit waters dramatically alter the binding energy landscape, improving the discrimination score from 0.27 to 0.89, and energetically favoring a structure only 0.43 Å from native. The *ECO* model predicts two water molecules that bridge the carboxyl group of the tyrosine ligand to interactions with an arginine side chain and a backbone nitrogen group (Fig 3C). Comparing the structure to the native crystal structure (at 2 Å resolution), we find that these two waters are 0.25 Å and 0.93 Å from native water positions; a third, more exposed water—also visible in Fig 3C— comes within 1.6 Å of a native water. Additional examples comparing recovered waters to crystal structures (PDB IDs 1N2J and 1U4D, at 1.8 Å and 2.1 Å resolution, respectively) are illustrated in panels 3E-H, illustrating four waters all within 1 Å from a crystallographic water position (see S11 Fig & S12 Fig for full solvated binding modes).

## Ligand docking scoring comparison

Finally, the new energy functions were compared against the results of a state-of-the art docking approach on a standardized dataset. A recent survey[22] of widely-used small molecules docking programs tested for performance against the Astex Diverse Set[23] which includes 85

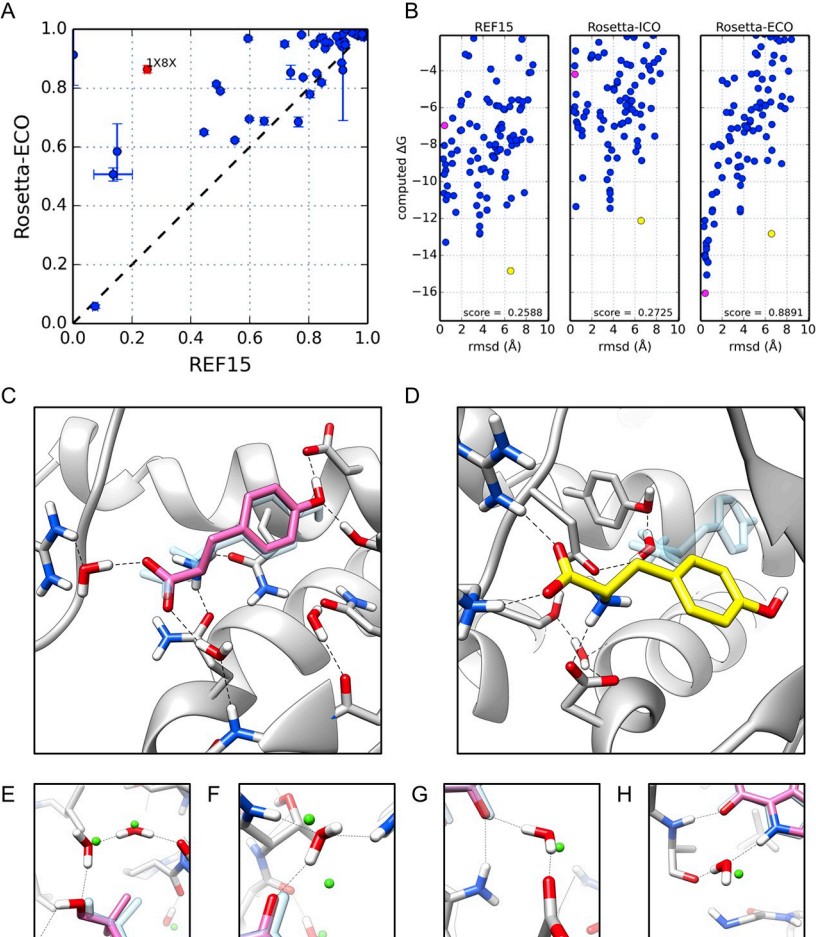

**Fig 3. Protein-ligand docking results. (A)** Scatter plot comparing results of 46 cases between baseline (*REF2015*) and *Rosetta-ECO*. Values are the Boltzmann-weighted discrimination score ± 1σ from an average of three independent runs. **(B)** Energy funnels, similar to Fig 2, for PDB ID 1X8X, tyrosyl t-RNA synthase bound to tyrosine (red data point in 3A) **C.** Explicitly-solvated, near-native docking pose in pink (RMSD = 0.43 Å; pink data point in 3B) with native ligand in transparent blue. **(D)** Explicitly-solvated decoy binding pose (RMSD = 6.57 Å; yellow data point in 3B). **(E-H)** A comparison of recovered waters (red) to high-resolution crystallographic waters (green spheres) from PDB ID: 1N2J (Panels E-G) and PDB ID: 1U4D (Panel H).

targets with ligands of pharmaceutical interest. We generated decoys for a 67-target subset (omitting cases where the ligand was additionally coordinated by an ion) using the docking software GOLD[24]. The GOLD-sampled structures were then rescored using the *REF2015*, *ICO*, and *ECO* energy functions of Rosetta. The results, fully presented in S1 Fig and S1 Table, show that while the Rosetta-rescored structures are more accurate than GOLD (78.2% versus 67.7% accuracy within a 1 Å RMSD cutoff; 94.6% versus 80.7% accuracy within 2 Å RMSD cutoff), little improvement is observed between *REF2015* and *ICO/ECO*. While these results suggest Rosetta may be a powerful tool for this dataset, the restricted conformational sampling obtained from GOLD (see S13 Fig for examples of sampling in RMSD space) does not benefit from the water model developments presented here and prevents a thorough evaluation of the energy functions. It is likely that a more evenly distributed set of docking conformations

would yield results similar to the score function improvements observed in the more tightly-curated protein/protein and protein/ligand data sets described above.

## Discussion

We have presented two approaches for considering coordinated water molecules in the prediction of native protein-protein and protein-ligand interfaces: *Rosetta-ICO*, which very efficiently captures the energetics of bridging waters implicitly, and *Rosetta-ECO*, which allows a small set of waters to emerge from bulk, resulting in a more physically complete representation of protein surfaces and interfaces. Both methods show improvements in protein interface recapitulation tasks with different levels of efficiency/accuracy tradeoffs: *Rosetta-ECO* more is accurate when it comes to decoy discrimination tests but 1.5–2 times slower than *Rosetta-ICO* depending on interface size. The level of native water recovery for *Rosetta-ECO* is about ~5% less than 3D-RISM for a similar precision level, yet the ECO model performs this task at ~10-fold increased speed while simultaneously predicting interface side chain configurations.

While the precision and recall reported by our explicit method might seem low, this is due to several factors. First, we are using a very strict recovery tolerance (0.5 Å, compared to 1.4–2.0 Å used elsewhere[9, 18]. Second, *Rosetta-ECO* is performing (and was designed to perform) a fundamentally different task than other approaches: simultaneously predicting both side chain geometry and coordinating waters. Nevertheless, our native water recovery numbers are encouraging when compared to a fixed-structure approach such as 3D-RISM, where results are similar when both methods are applied to the same PPI data set using the same recovery criteria.

Furthermore, while this work highlights the results of water prediction and protein interface recapitulation, we might expect the *Rosetta-ICO* energy function to show modest improvements at tasks related to monomeric structure prediction and protein sequence design. Indeed, that seems to be the case: when tested on independent datasets, modest improvements were observed in decoy discrimination with *ICO*. All other metrics were comparable between the two energy functions, leading us to conclude that the *ICO* model is a reasonable general-purpose energy function.

The improvement in both the protein and ligand docking tests suggests that these new energy functions may prove useful in the design of novel proteins intended to bind a particular ligand or protein. Successful design of protein-protein interfaces is often driven by van der Waals interactions that arise from shape complementarity, however better consideration of ordered solvent molecules may allow for the design of more natural interfaces which include numerous polar residues. Application of these new methods need not be limited to the solvation of interfaces or the description of binding partners. For example, the methods may be applied to more accurately predict the folded state of monomeric proteins in which buried solvent plays an important structural role or for prediction of the stabilizing or destabilizing effect of mutated residues on the surface of a protein. Additionally, the experiments described herein only consider the solvation of proteins and small molecules, however the framework can be easily extended to solvate other biomolecules such as nucleic acids.

## Methods

Two new biomolecular solvation methods are introduced here. The first (*Rosetta-ICO*) builds upon the existing implicit water model used in Rosetta to not only account for the energy of desolvating protein functional groups, but to additionally energetically favor conformations that are suitable to accommodate bridging waters. The second model (*Rosetta-ECO*) places

well-coordinated water molecules on the surface or at interfaces of biomolecules based largely on statistics from high-resolution experimental data.

## Implicit solvation (*Rosetta-ICO*)

An additional energy term is added to the Rosetta's implicit solvation model that models the energetic costs of highly ordered water molecules coordinated by multiple protein polar groups. The term builds upon our previously developed anisotropic solvation model[14], where for each polar group, one or more virtual water sites are placed in a configuration ideal for hydrogen bonding with the corresponding polar group. An energetic bonus is then given when the water sites of multiple polar groups overlap in such a way that a single water could coordinate, or "bridge", these polar groups:

$$E_{lk-bridge}(\boldsymbol{r}_i, \boldsymbol{r}_j)$$
$$= (E_{lk}^{(i,j)}) \cdot G(\max(\min_{\boldsymbol{w}_i, \boldsymbol{w}_j} \|\boldsymbol{w}_i - \boldsymbol{w}_j\| - D_{len}^0, 0); \ S_{len}^0) + (E_{lk}^{(i,j)})$$
$$\cdot G(\|\boldsymbol{b}_i - \boldsymbol{b}_j\| - D_{angle}^0; S_{angle}^0)$$

With:

$$G(x; S_0) = \begin{cases} \left(1 - \left(\dfrac{x^2}{S_0^{\ 2}}\right)^2\right)^2 & x \in [-S_0, S_0] \\ \\ 0 & x \notin [-S_0, S_0] \end{cases}$$

Here, $E_{lk}^{(i,j)}$ is the isotropic solvation term between atoms i and j (the fa_sol score term in the Rosetta energy function, see Fig 1A), $\boldsymbol{w}_i$ is the xyz coordinate of a theoretic water oxygen atom corresponding to polar group $\boldsymbol{r}_i$; $\boldsymbol{b}_i$ is the xyz coordinate of the base heavy atom used to construct the water (e.g., the backbone N or O), and $D_{len}^0$, $D_{angle}^0$, $S_{len}^0$, and $S_{angle}^0$ are parameters that are optimized during energy function evaluation, with final values of 0.5 Å, 4.33 Å, 1.61 Å, and 2.69 Å, respectively. Since a single polar atom may have multiple putative water binding sites, we take the minimum distance between all water sites corresponding to atoms i and j (the first term of the equation). Overall, the two terms in the equation characterize the overlap between potential water sites and the angle formed between polar groups that potentially coordinate a bridging water molecule.

This energy term was added to the current anisotropic solvation model in Rosetta (illustrated in Fig 1A–1D), and optimization of all polar terms was carried out (see S1 Text). Since this term does not prevent certain disallowed coordination geometries (e.g., 3 donors or 3 acceptors coordinating a single water site), we have introduced the *Rosetta-ECO* model to include fully modeled water molecules at possible hydration sites that can help filter out conformations with poor coordination geometry. Additionally, because this two-body energy term is only dependent upon the configuration of pairs of protein polar groups, it can be used in all Monte Carlo minimization methods used in Rosetta[25], with negligible computational overhead.

Additionally, to properly handle the geometry of water-protein and water-water hydrogen bonds, we modified the functional form of sp$_3$-hybridized hydrogen bond acceptors. Previously, the interaction between a hydrogen bond donor and the lone pair electrons of sp$_3$-hybridized acceptors was described by an angle and torsional term about the base atoms[26]; e.g., for serine, the angle CB-OG$\cdots$H$_{donor}$ and the pseudo-torsion HG-CB-OG$\cdots$H$_{donor}$. For water, however, this led to an undesirable property in that the potential was not symmetric about the two water hydrogens. Therefore, in *Rosetta-ICO* (and *Rosetta-ECO*) we replace the

torsional term for sp$_3$ hydrogen bond acceptors with a "softmax" potential between both atoms bonded to the sp$_3$-hybridized acceptor:

$$E_{sp3-chi}(\boldsymbol{a}_i, \boldsymbol{h}_j) = M \cdot \log\left(\sum_{\boldsymbol{b}_k \text{bound to} \boldsymbol{a}_i} \exp(E_{BAH}(\boldsymbol{b}_k, \boldsymbol{a}_i, \boldsymbol{h}_j)/M)\right)$$

Above, M describes the "softness" of the softmax with a default value of 0.4 kcal/mol (lower values make this function behave more like a "max"). The variables $\boldsymbol{b}_k$, $\boldsymbol{a}_i$ and $\boldsymbol{h}_j$ are the acceptor base atom, acceptor heavy-atom and donor hydrogen, respectively; and $E_{BAH}$ is the angular potential about the heavy-atom[26]. The summation is carried out over all bound atoms to the acceptor. For water acceptors, this would be over both hydrogens. In the serine example above, the angular potential is applied to both CB-OG···H$_{donor}$ and HG-OG···H$_{donor}$, with the softmax giving a score roughly equal to the worse of the two angular potentials. This ensures the potential is symmetric about both water hydrogens.

## Explicit solvation model (*Rosetta-ECO*)

One key challenge in prior explicit water modeling[27] is the large conformational space a single water molecule can adopt. This is an issue in applications (like those in this manuscript) where it is desirable to simultaneously sample side chain conformations and water positions. *Rosetta-ECO* makes use of a two-stage approach to navigate this problem (Fig 1E–1H). In the first stage, rotationally independent "point waters" are sampled using a statistical potential; not considering water rotation lets thousands of putative water positions be sampled efficiently. In the second stage, for the most favorable water positions (typically only several dozen) we consider rotations of these molecules using a physically derived potential.

In both steps of the protocol, Monte Carlo sampling is used to simultaneously sample side chain and water conformational states. In both stages, water molecules may be set to "bulk," losing an entropic penalty by doing so. This entropy bonus value, E$_{bulk}$, ultimately controls the number of explicit water molecules placed by the algorithm, requiring sufficient favorable physical interactions to overcome the entropic cost of coming out of bulk. This parameter was fit to a value of 1.22 kcal/mol. The atoms of any water molecule introduced into a model are subject to the same treatment by the full-atom Rosetta force field as any other atom, including interacting with bulk solvent via the lk_ball solvation model[14, 27]. Finally, rotational sampling of waters uses a uniform SO$_3$ gridding strategy[28] with 30° angular spacing, leading to 270 rotational conformers per water.

## Derivation of the statistical point water potential

The first step in determining possible water sites involves a low-resolution, statistical water potential to quickly evaluate the interaction between possible water sites and nearby polar groups of biomolecules. This potential, which we are calling the "point water potential", treats water molecules as simple, uncharged, points with attractive and repulsive Lennard-Jones terms.

The point water potential takes the form of:

$$E_{point-water}(W = \{\boldsymbol{w}_i, \ldots, \boldsymbol{w}_n\}) = \sum_{\text{waters } i} \sum_{\substack{\text{polar} \\ \text{atoms } j}} -\log P(\|\boldsymbol{w}_i, \boldsymbol{x}_i\|, \theta(\boldsymbol{w}_i, \boldsymbol{x}_j, \boldsymbol{x}_j^{base}))$$

$$-K \cdot \sum_{\substack{\text{waters } k \\ i \neq k}} \exp[-(\|\boldsymbol{w}_i, \boldsymbol{w}_k\| - 2.7)^2/\sigma^2] + E_{pwat\_bulk}$$

Here, $P$ is the statistical point-water distribution, parameterized over distance and angle; $d$ gives the distance between a water and polar atom, and $\theta$ gives the angle between the water position, the polar atom, and its "base atom." The point water energy term also considers other nearby point water sites, k, as Gaussian distributions with width $\sigma$ and height $K$ (with min energy at a distance of 2.7 Å), which was determined by averaging water-water distances observed in high resolution crystal structures. K and σ were optimized, yielding values of 0.52 kcal/mol and 0.24 Å, respectively. Finally, an overall energetic cost of bringing the water molecule "out of bulk," $E_{pwat\_bulk}$, is added for each water, with a value of 2.71 kcal/mol. These parameters were fit using crystallographic waters in the Top8000 database (see Supporting Information for more details).

## Identifying and sampling point waters positions

A key challenging in building possible water sites is the desire to simultaneously sample side chain conformations along with water positions. Thus, the initial placement of water molecules to be optimized by the point water potential come from two sources: a) ideal solvation about protein backbones and b) *possible* solvation sites from side chain rotamers. For backbone waters, point generation is straightforward: 1 "ideal" site for each backbone N-H group and 10 "ideal" sites are generated from each backbone C = O group (based on clustering waters from crystal structures, S15 Fig & S16 Fig).

Generation of side chain-coordinated waters is more involved. Considering all possible water molecules that may coordinate the polar groups of all side chain rotamers leads to water conformer sets that are unmanageably large to sample. Thus, we again build off prior work[29] and consider instead the overlapping hydration sites that emanate from two different side chain or backbone groups. That is, we collect the idealized hydration sites for all possible side chain rotamers and identify all positions where there is overlap (within 0.75 Å) between two potential water sites originating from different side chains or backbone groups. A 3D hash table makes this calculation efficient even when there are millions of putative water positions. Finally, to further reduce conformational sampling, during the Monte Carlo "packing" algorithm, when both side chain and point water positions are sampled, all putative point waters are clustered into sets in which only one site can be occupied.

A modified version of Rosetta's traditional packing algorithm[30] is used when point waters are present. Typically, Rosetta uses simulated annealing to find the discrete rotamer set minimizing system energy, where the temperature of the trajectory is slowly annealed from RT = 100 to RT = 0.3 kcal/mol. With the point water potential, we do not expect the force field (which does not consider water rotation) to be perfect, and we want the packer not to optimize total energy but to simply separate reasonable from unreasonable water positions for a more expensive subsequent calculation. Thus, we instead used long simulations at low temperatures (RT = 0.3) at which the "dwell time" of each state is recorded, with intervening high-temperature "spikes" (RT = 100) used to periodically scramble the state which may settle into various low-energy minima of the potential energy surface. Then, instead of taking the lowest energy state sampled, we measured water "occupancy" at each position, taking point water positions with a "dwell time" greater than 2% (ignoring occupancy counts during the high-temperature steps and the first 1/6 of low-temperature steps of each iteration).

Water positions passing this criterion, typically on the order of dozens to hundreds, are then filled with three-point water molecules which are allowed to rotate about fixed oxygen positions and are sampled (along with all surrounding side chains) using Rosetta's standard simulated annealing rotamer optimization routine. The Monte Carlo algorithm is unaltered from the standard packing routine in Rosetta, in that a random rotamer (side chain or water) is selected and tested against a Metropolis criterion. The only exception here is that when a water rotamer, which is a rotational state about a fixed oxygen position, is selected for sampling, there is a 50% chance that the "virtual" state/rotamer of the water molecule is sampled instead. Given that the first stage in the solvation routine places a substantially larger than expected population of water sites on the surface of the biomolecule, a majority of these sites will not result in water conformations that are well-coordinated by the surrounding protein or ligand polar atoms. This adjustment to the sampling of water states helps with the convergence of the sampling problem with so many potential false positive water molecules.

Finally, during subsequent minimization of the water-containing model, the kinematics used to minimize water molecules (the fold tree) in Rosetta optimizes 6 degrees of freedom for each water, representing the rigid-body transformation between the water and nearest amino-acid (using Rosetta terminology, the "jump" is defined between the nearest amino acid and the water).

## Datasets

Four different data sets were used in the testing of the new energy functions described here. The first includes 153 high-resolution crystal structures of protein-protein interfaces (PPIs) that was used for both native water and rotamer recovery at the interfaces. Two docking data sets were used to test the ability of the new energy functions to discriminate near-native from decoy docking conformations, a subsets of those used by Park et al.[14], but selected for water-rich interfaces (and to exclude problematic cases such as PPIs with disulfides across the interface or ions contributing to binding). For protein-protein interactions, a 53-case subset of the ZDock 4 Benchmark set[31] was used, while a 46-case subset of the Binding MOAD database [32] was used for protein-ligand interactions. Finally, another ligand docking set, generated with GOLD on a subset of the Astex Diverse Set[23] was used to compare the new energy functions against an established docking score function. All conformational sampling to generate the docking datasets was performed with fixed protein backbones. There is no overlap between the datasets used for parameter training and those used for the docking discrimination tests, and while there is significant overlap between the protein-ligand and Astex sets, these were used for different purposes and with significantly different sampling strategies. Additional details on the datasets, including lists of PDB IDs used are included in the Supporting Materials.

## Benchmarking against 3D-RISM water site predictions

The water site predictions in Rosetta were compared against those predicted by the 3D-RISM method[33] as implemented in AmberTools19[16, 34]. Briefly, RISM calculations were performed for pure water at a concentration of 55.5 M with a 0.5 Å grid spacing. Using a buffer of 7 Å, as opposed to the default 14 Å, was found to be speed up calculations while not hurting recovery for our dataset which consists of water molecules found at PPIs. The Placevent algorithm[35] was used to determine explicit water sites, which were truncated to be found within 6 Å of all CB atoms (CA for GLY) of the residues that form the interfaces of the test set. This was done to be comparable to the *Rosetta-ECO* results, in which water sampling was limited to protein/protein interfaces. Finally, the results were further trimmed by the 3D-RISM water-

protein radial distribution function (RDF > = 10.2) to achieve the same level of precision as *Rosetta-ECO*.

## Binding energy calculations

The binding energies, $\Delta G_{bind}$, were calculated for the near-native and incorrect (decoy) docking poses by taking the difference between the computed energies of the bound and unbound states. This is accomplished in Rosetta by first calculating the energy for the bound system, then re-computing the energy when the two binding components are separated to obtain unbound state energies. An important part of interface energetics involves computing the energy cost of water displacement[36], making treatment of explicit waters of the unbound state an important consideration. Due to size differences of the average interface, we found slightly different treatment performed better with PPIs versus protein-ligand interfaces. In both PPIs and protein-ligand interfaces, the bound states are solvated (including reoptimization of interface side chains), using the two-stage Monte Carlo procedure described above, restricting water placement to only the biomolecular interface of interest. Given that this mode of solvation samples both side chain and water orientations, our strategy considers the induced fit effect on a fixed backbone level. Then all side chains are minimized and, for protein-ligand interfaces only with the *ICO* model, the rigid-body transformation between receptor and ligand is also minimized. Interface components are then separated and re-solvated. Copies of the waters from the bound state are duplicated such that one copy belongs to both ligand and receptor, while the re-solvation protocol restricts new water placement to the same region that defined the interface in the bound state. During the resampling of the unbound state, side chains that previously defined the interface are once again reoptimized, allowing waters that were previously highly coordinated in the bound state to be liberated to bulk if a sufficient part of this coordination was lost in the unbinding process. Any water molecules that remain unliberated to bulk following sampling are considered part of the bound/unbound states for scoring purposes.

RMSD values reported for docking are of the small molecule or protein ligand with respect to the native experimental structure. Ligand C$\alpha$ RMSDs are used for protein-protein docking cases, where the ligand is the second chain in the experimental PDB file, while heavy atom RMSDs are used for small molecule docking cases. Sample XML scripts used for the protein/ protein and protein/ligand rescoring are included in the Supporting Information.

## Training tasks

The training tasks used for energy function parameterization are the same as detailed in the development of the REF2015 Rosetta energy function[14] and are summarized in the Supporting Information.

## Supporting information

**S1 Data. Data Set 1.** PDB files used for water recovery tests. Protein-Protein (16 GB) and Protein-Ligand (1.6 GB) decoys sets available at https://github.com/rpavlovicz/rpavlovicz-docking_data_sets.
(DOCX)

**S1 Fig. Rescoring GOLD docking results with Rosetta.** Results for rescoring Astex Diverse Set. Docking conformations initially generated and scored by GOLD (red) were rescored with the Rosetta *REF2015* energy function (blue). The theoretical scoring success is determined by the initial GOLD sampling (black dashed) for the 67 cases of the Astex Diverse Set that do not

coordinate an ion in the binding site.
(PNG)

**S2 Fig. Protein-protein docking scoring results (part 1).** Recalculation of protein-protein docking interface scores ($\Delta G_{bind}$) for three different Rosetta scoring functions: REF2015, *Rosetta-ICO*, and *Rosetta-ECO*. Data points represent the average of three runs with the standard deviation as error bars. The average Boltzmann discrimination scores +/- standard deviation for each distribution is found in the bottom right corner of each plot.
(TIF)

**S3 Fig. Protein-protein docking scoring results (part 2).** Recalculation of protein-protein docking interface scores ($\Delta G_{bind}$) for three different Rosetta scoring functions: REF2015, *Rosetta-ICO*, and *Rosetta-ECO*. Data points represent the average of three runs with the standard deviation as error bars. The average Boltzmann discrimination scores +/- standard deviation for each distribution is found in the bottom right corner of each plot.
(TIF)

**S4 Fig. Protein-protein docking scoring results (part 3).** Recalculation of protein-protein docking interface scores ($\Delta G_{bind}$) for three different Rosetta scoring functions: REF2015, *Rosetta-ICO*, and *Rosetta-ECO*. Data points represent the average of three runs with the standard deviation as error bars. The average Boltzmann discrimination scores +/- standard deviation for each distribution is found in the bottom right corner of each plot.
(TIF)

**S5 Fig. Protein-protein docking scoring results (part 4).** Recalculation of protein-protein docking interface scores ($\Delta G_{bind}$) for three different Rosetta scoring functions: REF2015, *Rosetta-ICO*, and *Rosetta-ECO*. Data points represent the average of three runs with the standard deviation as error bars. The average Boltzmann discrimination scores +/- standard deviation for each distribution is found in the bottom right corner of each plot.
(TIF)

**S6 Fig. Protein-protein docking scoring results (part 5).** Recalculation of protein-protein docking interface scores ($\Delta G_{bind}$) for three different Rosetta scoring functions: REF2015, *Rosetta-ICO*, and *Rosetta-ECO*. Data points represent the average of three runs with the standard deviation as error bars. The average Boltzmann discrimination scores +/- standard deviation for each distribution is found in the bottom right corner of each plot.
(TIF)

**S7 Fig. Protein-ligand docking scoring results (part 1).** Recalculation of protein-ligand docking interface scores ($\Delta G_{bind}$) for three different Rosetta scoring functions: REF2015, *Rosetta-ICO*, and *Rosetta-ECO*. Data points represent the average of three runs with the standard deviation as error bars. The average Boltzmann discrimination scores +/- standard deviation for each distribution is found in the bottom right corner of each plot.
(TIF)

**S8 Fig. Protein-ligand docking scoring results (part 2).** Recalculation of protein-ligand docking interface scores ($\Delta G_{bind}$) for three different Rosetta scoring functions: REF2015, *Rosetta-ICO*, and *Rosetta-ECO*. Data points represent the average of three runs with the standard deviation as error bars. The average Boltzmann discrimination scores +/- standard deviation for each distribution is found in the bottom right corner of each plot.
(TIF)

**S9 Fig. Protein-ligand docking scoring results (part 3).** Recalculation of protein-ligand docking interface scores ($\Delta G_{bind}$) for three different Rosetta scoring functions: REF2015, *Rosetta-ICO*, and *Rosetta-ECO*. Data points represent the average of three runs with the standard deviation as error bars. The average Boltzmann discrimination scores +/- standard deviation for each distribution is found in the bottom right corner of each plot.
(TIF)

**S10 Fig. Protein-ligand docking scoring results (part 4).** Recalculation of protein-ligand docking interface scores ($\Delta G_{bind}$) for three different Rosetta scoring functions: REF2015, *Rosetta-ICO*, and *Rosetta-ECO*. Data points represent the average of three runs with the standard deviation as error bars. The average Boltzmann discrimination scores +/- standard deviation for each distribution is found in the bottom right corner of each plot.
(TIF)

**S11 Fig. 1N2J binding mode with *Rosetta-ECO* model.** The near native *Rosetta-ECO* model is in thicker stick representation with full-atom water molecules and the ligand depicted in pink. The experimental ligand (pantoate) position is in transparent blue, water oxygen positions as green spheres, and native side chains are in black wire representation. If the native ligand or side chain positions cannot be seen, it is because they are obscured by the Rosetta model. Panel A highlights the overall binding pocket, while panels B-D focus on recovered water positions.
(TIF)

**S12 Fig. 1U4D binding mode with *Rosetta-ECO* model.** The near native *Rosetta-ECO* model is in thicker stick representation with full-atom water molecules and the ligand depicted in pink. The experimental ligand (debromohymenialdisine) position is in transparent blue, water oxygen positions as green spheres, and native side chains are in black wire representation. If the native ligand or side chain positions cannot be seen, it is because they are obscured by the Rosetta model. Panel A highlights the overall binding pocket, while panels B and C focus on recovered water positions.
(TIF)

**S13 Fig. Comparison of docking scores/energies for conformations sampled By GOLD for select cases.** The RMSD of the ligand from the experimental conformation is plotted against the computed score (ChemPLP) for GOLD and $\Delta G_{bind}$ for Rosetta. Note that the sampling from GOLD is often focused in small number of docking conformations, leaving gaps in the sampled space.
(TIF)

**S14 Fig. Derivation of a statistical water potential. Upper left:** Distribution of waters about histidine residues over a range of distance from the HD1 atom and a range of angles from the HD1 and ND1 atoms [-log($HIS_{HD1\_ND1}$)] **Upper right:** Distribution of waters about a non-polar reference [log($ALA_{HB1\_CB1}$)] **Lower left:** The sum of the upper two figures: the statistical potential for histidine **Lower right:** Final, modified histidine potential filtered for noise and second solvation shell effects.
(PNG)

**S15 Fig. Sample statistics of waters about peptide C = O groups. Upper right:** distance and angle of all waters measured (grey) and those used for statistical placement about the polar group (purple). **Bottom left:** Angle and dihedral distribution with histogram projections in upper left (angle) and lower right (dihedral).
(PNG)

**S16 Fig. Position of cluster representatives for solvation of C = O backbone groups.** The crystallographic water positions used for statistical placement of potential solvation sites about C = O backbone polar groups are shown here in red, with the k-means cluster centroids (k = 10) illustrated in yellow. Two views of these data are shown about an arbitrary alanine residue.
(PNG)

**S17 Fig. Rotamer recovery error as a function of native water positions randomly perturbed.** Crystallographic water molecules in our benchmark set were randomly perturbed 0.0 to 1.6 Å and the interface residues were repacked in Rosetta. Data points represent the average of three independent runs with 95% confidence interval error bars. The baseline of packing the interfaces without any water molecules (REF2015 score function) is shown as a dashed grey line with 95% confidence intervals from three runs shaded in light grey.
(PNG)

**S1 Table. GOLD docking and Rosetta rescoring results of Astex Diverse Set.**
(DOCX)

**S2 Table. 3D-RISM results on interface water test set.**
(DOCX)

**S3 Table. Timing comparison between *Rosetta-ECO* and 3D-RISM.**
(DOCX)

**S1 Text. Rosetta force field parameters.** Final parameters used in *Rosetta-ICO* and *ECO* force fields.
(DOCX)

**S2 Text. Dataset information.** Details on datasets used for water recovery and docking discrimination tests, including GOLD docking protocol.
(DOCX)

**S3 Text. Additional information.** Includes information about CAPRI Target 47, details on the derivation of low-resolution statistical water potential, and more information on the force field parameter training tasks.
(DOCX)

**S1 Script. XML script.** RosettaScripts XML file used for protein-protein / protein-ligand interface scoring with explicit water molecules (*Rosetta-ECO*).
(DOCX)

## Acknowledgments

This work was facilitated though the use of advanced computational, storage, and networking infrastructure provided by the Hyak supercomputer system at the University of Washington. Structure visualization and analysis used the UCSF Chimera software[37], while GNU Parallel was used for distributed processing and data analysis[38].

## Author Contributions

**Conceptualization:** Ryan E. Pavlovicz, Hahnbeom Park.

**Methodology:** Ryan E. Pavlovicz, Hahnbeom Park.

**Software:** Ryan E. Pavlovicz, Hahnbeom Park.

**Supervision:** Frank DiMaio.

**Writing – original draft:** Ryan E. Pavlovicz.

**Writing – review & editing:** Hahnbeom Park, Frank DiMaio.

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
