## [Decision Letter · Decision Letter 0]

30 Jan 2020

Dear Dr. DiMaio,

Thank you very much for submitting your manuscript "Efficient consideration of coordinated water molecules improves computational protein-protein and protein-ligand docking discrimination" for consideration at PLOS Computational Biology.

As with all papers reviewed by the journal, your manuscript was reviewed by members of the editorial board and by several independent reviewers. In light of the reviews (below this email), we would like to invite the resubmission of a significantly-revised version that takes into account the reviewers' comments.

We cannot make any decision about publication until we have seen the revised manuscript and your response to the reviewers' comments. Your revised manuscript is also likely to be sent to reviewers for further evaluation.

Sincerely,

Björn Wallner

Associate Editor

PLOS Computational Biology

Mona Singh

Methods Editor

PLOS Computational Biology

Reviewer's Responses to Questions

**Comments to the Authors:**

Reviewer #1: Most protein structure prediction, docking and design methods use implicit solvent for efficiency, but it is well known that specific waters can appear in crystallographic interfaces and folds. One way to keep the efficiency of implicit models is to include only a few explicit waters in a hybrid model, but this approach proved too slow in prior attempts. Here, this paper presents two new methods to model coordinated waters in biomolecular structures. The first, ICO, is still implicit but examines overlap of virtual water placement sites. The second, ECO, is semi-explicit and places three-atom waters using hash tables and Monte Carlo approaches to keep it fast. The results show that it is still quite difficult to place crystallographic waters (under 18% recovered with under 18% precision), but nevertheless this reduces side-chain placement error modestly. More importantly, it clearly improves discrimination of protein-protein and protein-ligand docking poses upon refinement, showing that, as has been conjectured, the lack of explicit waters at interfaces is one of the limiting factors in docking. The paper marks a significant advance in methods and should be published. I have a few small points on the main paper, and several clarifications needed in the methods. My long list of methods questions is to help understand this calculation and is not an indication of shortcomings of the main work.

Main text / major comments:

1. The CAPRI challenge included a water-placement target a few years ago. It would be interesting to see how these methods could perform on that target, and it would provide a clear metric of these methods against a large set of methods in the field. See https://www.ncbi.nlm.nih.gov/pmc/articles/PMC4582081/ for details; at CAPRI meetings, Lensink has even offered to provide the evaluation scripts as needed.

2. Please add a couple more sentences to the main text describing the main hypotheses and concepts behind the two models.

3. In Figure 1A-D, the coordinate system is not clear. What are we looking at here?

4. Binding energy: are the separated components repacked or relaxed? I think not from the methods, please specify (line 470). (Chaudhury found that not relaxing shows a clearer signal for docking; he called it “interface energy” to distinguish it from the relaxed-monomer case which is more like nature.) Also are waters in the separated components placed all over the monomers or just at the binding interface? It was nice that the method buries about the same number of waters at interfaces as Janin counted in crystals; can you also compare the waters at the separated interfaces and are they also like Janin or another reference of structured waters?

5. It seems like the docking is done with a rigid protein backbone. Please confirm and make sure that is clear in the paper. Or, if there is backbone motion, describe and explain its impact on the results.

Methods questions:

6. Table 2: which rmsd measure is used? Lrmsd? Irmsd? Atom selection? Please define in methods.

7. Line 176: In the benchmark set, the 53 protein-protein interfaces are categorized as rigid, moderate or flexible backbone targets. Which were chosen? Is Rosetta-ICO and Rosetta-ECO more or less helpful on any of these categories?

8. Line 177: As per line 466 and comment 4 above, use “interface energy”

9. Line 339: Add units on D0 and S0. I suggest replacing S0 with S0^2 in the formula so that both parameters can have dimensions of length. This would help make it physically interpretable.

10. Line 335: Why was the form of G chosen? It seems like it might be very sharp at the minimum. Add a plot with the final values of D0 and S0 and units.

11. What is the reference energy for the lk-bridge term? As atoms move away, the energy goes to infinity! How is this handled?

12. Line 340-341: I get how the second term targets for distance, but the first term seems to look to superpose the two water molecules. I guess ‘angle’ comes out of that, but I suggest rephrasing this sentence.

13. Line 343: Please add a table of all the new parameters, to compare with the one in Alford et al and show the differences from the old set. Also, in the text, please give the nominal values for the new fitted parameters S0 and D0.

14. Line 346: I’m not sure what ‘quite reasonable’ means. Change to state the fraction of time that a donor or acceptor violates its number of physically reasonable bonds.

15. Line 348: citation doesn’t seem right.

16. Line 355: it took me a long time to figure out what was meant by treating the ‘water asymmetrically’. Maybe add another example (in addition to the serine example) with the water atoms as an example. If I’m following correctly, the water will not even have a pseudo-torsion energy, correct?

17. Line 355: what is M? Please define with final value and units. Also I think the sign might be wrong here.

18. E_ref is a poor name for the energy of releasing a water to the bulk because this symbol is already used in Rosetta many times. How about E_bulk or E_water_release or E_WS for water/entropy…anything that is more directly tied to its meaning here.

19. I wasn’t sure the “lk” suffix is useful on the bridge energy term name. Sure, it’s based on the Lazaridis-Karplus base model, but this is a whole new water approach. Is this just so it sorts in an alphabetical list near the other solvation terms? Maybe beyond the scope here, but maybe all solvation terms should be renamed E_solv_*, where * is lk or otherwise.

20. Is the lk_ball model used in ICO or ECO?

21. Line 391, what is the capital W about? Should that be a script W to denote the set of waters placed?

22. Line 391, replace d() with a norm function as in the earlier equation.

23. Line 394 add a small picture; I presume the angle is between w, a and b?

24. Line 397 give the final numbers for sigma and K with units.

25. Line 410, why not nitrogen backbone H-bonding sites too? Are side chain sites enumerated only from oxygens, not nitrogens?

26. Paragraph starting line 411, I don’t quite get the meaning and how this calculation is done. If rot_i1 clashes with rot_j1 but does not clash with rot_j2, what happens? When you detect overlaps in line 414 it sounds like it’s about sc-sc overlaps, but line 417 seems to talk about w-sc overlaps too? Please clarify as this hash table approach seems to be a key to speeding the calculation relative to the older solvated rotamers approaches. What’s in the hash table? Water positions?

27. For calculating ‘dwell time,’ is that only at the low temperature? After an equilibration period each time?

28. For the ECO monte carlo, please specify the MC move set and the fractional probability of calling each move type and the sizes of the moves (e.g. translations and rotations). The fold tree seems important too…do the waters move with one or both docking partners? Maybe all molecules (protein, ligand, and water) tie to a virtual origin point to move independently?

29. Line 466: Binding energies imply a transition from the unbound form of receptors and ligands to the bound form of the complex. The energy calculations described involve the energy of bound conformations of receptors and ligands as they form the binding complex, and would be better defined as the “interface” (or interaction) energy

Grammatical edits

1. In all the equations, some terms are xyz vectors and some are scalars. Consider using boldface for the xyz vectors to make them easier to interpret.

2. Line 142, add “two” before “predicted water sites”

3. Line 128: “implemented”

4. Line 224: change “scores” to “discrimination scores”? Also in Supp. Figure captions.

5. Line 319: not only account for desolvation penalties, but “also” energetically reward ….

6. Line 355 I think “replaced” should be “was replaced”?

7. Line 405: “conformations”

8. Line 414 “challenge”

9. Line 424 add “kcal/mol” units to RT

10. Line 459 capitalize “Placevent”

11. Line 444 should probably cite the separate Weng paper that just presents the docking benchmark

Reviewer #2: The manuscript by Pavlovicz et al. focuses on prediction of water molecules in protein-protein and protein-ligand complexes by implementing new methods in the Rosetta program. An impressive and original feature of the approach is that water positions and side chain rotamers are sampled at the same time, whereas most previous approaches have used a rigid protein. The authors also evaluate the performance of predictions of protein-protein and protein-ligand complexes. The research topic is relevant and important – waters are important for molecular recognition and accurate positioning of water should improve the performance of scoring functions. The results are mixed. In some cases, an improvement is observed with the waters included, in other cases it has no effect. I think that further analysis of the results is needed. I recommend that the following changes are made to the manuscript:

(1) Page 3: The authors write that their approach yields “superior results” at the end of the introduction. Compared to what?

(2) Page 4: I was surprised to read that only 17.7% of the native water molecules were predicted by the method for protein-protein interfaces. With previous methods (e.g. https://journals.plos.org/plosone/article?id=10.1371/journal.pone.0032036, https://pubs.acs.org/doi/10.1021/ci200150p) that are likely as fast or faster than the presented one, 76-97% of waters in crystal structures are predicted. These studies focus on proteins (not protein-protein complexes) and likely use rigid side chains, but can that explain the substantially lower performance of the method presented in this work? Or is there some other difference in the assessments (I think there might be some)? I recommend that the authors validate their method in the same way as these previous studies (e.g. with static side chains and same cutoffs) and comment on potential differences in the results. This analysis should not only include protein-protein complexes, but also protein-ligand complexes. This will give the reader a better idea of the performance.

(3) Page 7: The authors compare they results to 3D-RISM, which gives better results, but is much slower (20-fold). However, there are many very fast methods too, which have not been mentioned (see previous comment). The relative performance of the methods and their advantages/disadvantages should also be mentioned in the discussion.

(4) Page 9: The improvement of docking performance for Rosetta-ECO is very encouraging. However, the low prediction accuracy for waters (page 4) makes me question if the results are “right for the right reason”. The authors should quantify if the top-ranked complexes for which an improvement is observed also has accurately placed waters compared those observed in the crystal structure.

(5) Related to the previous question, in Figure 3C-D: Can you compare to the crystal structure water positions and side chain conformations in this case? How many of the water predictions are confirmed? Inspection of 1X8X revealed a phosphate close to the ligand.

**Have all data underlying the figures and results presented in the manuscript been provided?**

Reviewer #1: No: Some parameter values requested to add in the methods. (software is available)

Reviewer #2: Yes

PLOS authors have the option to publish the peer review history of their article (what does this mean?). If published, this will include your full peer review and any attached files.

Reviewer #1: Yes: Jeffrey J. Gray and a student

Reviewer #2: No
---

## [Decision Letter · Decision Letter 1]

8 Jun 2020

Dear Dr. DiMaio,

Thank you very much for submitting your manuscript "Efficient consideration of coordinated water molecules improves computational protein-protein and protein-ligand docking discrimination" for consideration at PLOS Computational Biology. As with all papers reviewed by the journal, your manuscript was reviewed by members of the editorial board and by several independent reviewers. The reviewers appreciated the attention to an important topic. Based on the reviews, we are likely to accept this manuscript for publication, providing that you modify the manuscript according to the review recommendations.

Sincerely,

Björn Wallner

Associate Editor

PLOS Computational Biology

Jason Papin

Editor-in-Chief

PLOS Computational Biology

[LINK]

Reviewer's Responses to Questions

**Comments to the Authors:**

Reviewer #1: Thank you for the CAPRI target analysis. This is quite nice. I think you should include it in the paper, maybe even summarize the result in the abstract, because it places your work within the context of the field. At a minimum, put the text in your reviewer response in the Supplement.

All other concerns are well addressed in the text. Thank you for being so careful with the theory, methodological details, and presentation.

The method is very exciting and is sure to be used by many, teaching us new things about the role of specific waters in structures.

Reviewer #2: The authors have addressed most of my questions, and only a minor addition is required.

I am puzzled about why the authors think that comparison to simplified methods (e.g. WaterDock, WaterMap, AcquaAlta) is not so relevant. I think the authors should extend the clarifying section about this in the discussion and explicitly mention the results (e.g. % reproduced water) achieved by such methods. The difference is interesting for the reader and may influence the choice of method to apply in applications. The authors could provide some advice on this also based on the results.

**Have all data underlying the figures and results presented in the manuscript been provided?**

Reviewer #1: Yes

Reviewer #2: Yes

PLOS authors have the option to publish the peer review history of their article (what does this mean?). If published, this will include your full peer review and any attached files.

Reviewer #1: Yes: Jeffrey J. Gray & Ameya Harmalkar

Reviewer #2: No
---

## [Decision Letter · Decision Letter 2]

29 Jun 2020

Dear Dr. DiMaio,

We are pleased to inform you that your manuscript 'Efficient consideration of coordinated water molecules improves computational protein-protein and protein-ligand docking discrimination' has been provisionally accepted for publication in PLOS Computational Biology.

Best regards,

Björn Wallner

Associate Editor

PLOS Computational Biology

Mona Singh

Methods Editor

PLOS Computational Biology

Reviewer's Responses to Questions

**Comments to the Authors:**

Reviewer #1: Thank you for including the CAPRI analysis in the paper, it will be a nice reference for others.

Reviewer #2: My points have been addressed and I recommend acceptance.

**Have all data underlying the figures and results presented in the manuscript been provided?**

Reviewer #1: Yes

Reviewer #2: Yes

PLOS authors have the option to publish the peer review history of their article (what does this mean?). If published, this will include your full peer review and any attached files.

Reviewer #1: **Yes: **Jeffrey J. Gray

Reviewer #2: No

---

## [Editor Report · Acceptance letter]

28 Aug 2020

PCOMPBIOL-D-19-01956R2 

Efficient consideration of coordinated water molecules improves computational protein-protein and protein-ligand docking discrimination

Dear Dr DiMaio,

I am pleased to inform you that your manuscript has been formally accepted for publication in PLOS Computational Biology. Your manuscript is now with our production department and you will be notified of the publication date in due course.

With kind regards,

Matt Lyles
